# POLICY IMPROVEMENT BY PLANNING WITH GUMBEL

**Ivo Danihelka**[1,2]**, Arthur Guez**[1]**, Julian Schrittwieser**[1]**, David Silver**[1,2]
[1]DeepMind, London, UK
[2]University College London
`danihelka@google.com`

## ABSTRACT

AlphaZero is a powerful reinforcement learning algorithm based on approximate policy iteration and tree search. However, AlphaZero can fail to improve its policy network, if not visiting all actions at the root of a search tree. To address this issue, we propose a policy improvement algorithm based on sampling actions without replacement. Furthermore, we use the idea of policy improvement to replace the more heuristic mechanisms by which AlphaZero selects and uses actions, both at root nodes and at non-root nodes. Our new algorithms, Gumbel AlphaZero and Gumbel MuZero, respectively without and with model-learning, match the state of the art on Go, chess, and Atari, and significantly improve prior performance when planning with few simulations.

## 1 INTRODUCTION

In 2018, AlphaZero (Silver et al., 2018) demonstrated a single algorithm achieving state-of-the-art results on Go, chess, and Shogi. The community reacted quickly. Leela Chess Zero (Linscott et al., 2018) was created to reproduce AlphaZero results on chess, winning Top Chess Engine Championship in 2019. Soon, all top-rated classical chess engines replaced traditional evaluations functions with Efficiently Updatable Neural Network (Nasu, 2018).

AlphaZero was itself generalized by MuZero (Schrittwieser et al., 2020). While AlphaZero requires a black-box model of the environment, MuZero learns an abstract model of the environment. Essentially, MuZero learns the rules of Go, chess, and Shogi from interactions with the environment. This allows MuZero to excel also at Atari and continuous control from pixels (Hubert et al., 2021).

In this work, we redesign and improve AlphaZero. In particular, we consider the mechanisms by which AlphaZero selects and uses actions, which are based upon a variety of heuristic ideas that have proven especially effective in Go, chess, and Atari (Silver et al., 2018; Schrittwieser et al., 2020). However when using a small number of simulations, some of AlphaZero's mechanisms perform poorly. We use the principle of policy improvement to suggest new mechanisms with a better theoretical foundation. More specifically, we consider each mechanism in turn, alongside our proposed modifications:

- **Selecting actions to search at the root node.** To explore different actions during training, AlphaZero selects actions by adding Dirichlet noise to its policy network, and then performs a search using the perturbed policy. However, this does not ensure a policy improvement. We instead propose to sample actions without replacement by using the Gumbel-Top-k trick (Section 2), and perform a search using the same Gumbel values to influence the selection of the best action (Section 3.3), and show that this guarantees a policy improvement when action-values are correctly evaluated.

- **Selecting actions at the root node.** AlphaZero uses a variant of the PUCB algorithm (Rosin, 2011) to select actions at the root node. This algorithm was designed to optimize cumulative regret in a *bandit-with-predictor* setting (i.e. given prior recommendations from the policy network). However, no ancestors are dependent upon the evaluation of the root node, and the performance of the Monte-Carlo tree search therefore only depends upon the final recommended action at the root node, and not upon the intermediate actions selected during search (Bubeck et al., 2011). Consequently, we propose to use the Sequential Halving algorithm (Karnin et al., 2013) at the root node to optimize simple regret in a stochastic bandit with a predictor (Section 3.4).

- **Selecting actions in the environment.** Once search is complete, AlphaZero selects an action by sampling from an (annealed) categorical distribution based upon the visit counts of root actions resulting from the search procedure. We instead propose to select the singular action resulting from the Sequential Halving search procedure.

- **Policy network update.** AlphaZero updates its policy network towards a categorical distribution based upon the visit counts of root actions. However, even if the considered actions are correctly evaluated, this does not guarantee a policy improvement, especially when using small numbers of simulations (Grill et al., 2020). We instead propose a policy improvement based upon the root action values computed during search, and update the policy network towards that policy improvement (Section 4).

- **Selecting actions at non-root nodes.** AlphaZero uses the PUCT algorithm to select actions at non-root nodes. We instead propose to select actions according to a policy improvement (similar to the proposal of Grill et al. (2020)) based upon a completion of the action values. Furthermore, rather than sampling directly from this policy improvement, we propose a deterministic action selection procedure that matches the empirical visit counts to the desired policy improvement (Section 5).

The proposed modifications are applicable also to MuZero or any agent with a policy network and an expensive Q-network. The modifications are most helpful when using a small number of simulations, relative to the number of actions. When using a large number of simulations, AlphaZero works well. We tried to ensure that the new search is principled, better with a smaller number of simulations, and never worse. We succeeded on all tested domains: Go, chess, and Atari.

## 2 BACKGROUND

Before explaining the improved search, we will explain the Gumbel-Max trick and the Gumbel-Top-k trick. The Gumbel-Max trick was popularized by Gumbel-Softmax for a gradient approximation. In this paper, we are not interested in approximate gradients. Instead, we use the Gumbel-Top-k trick to sample without replacement.

**Gumbel-Max trick.** (Gumbel, 1954; Luce, 1959; Maddison et al., 2017; Jang et al., 2017)
Let $\pi$ be a categorical distribution with $\mathrm{logits} \in \mathbb{R}^k$, such that $\mathrm{logits}(a)$ is the logit of the action $a$. We can obtain a sample $A$ from the distribution $\pi$ by first generating a vector of $k$ Gumbel variables and then taking argmax:

$$(g \in \mathbb{R}^k) \sim \mathrm{Gumbel}(0) \tag{1}$$
$$A = \arg\max_a (g(a) + \mathrm{logits}(a)). \tag{2}$$

**Gumbel-Top-k trick.** (Yellott, 1977; Vieira, 2014; Kool et al., 2019) The Gumbel-Max trick can be generalized to sampling $n$ actions *without replacement*, by taking $n$ top actions:

$$(g \in \mathbb{R}^k) \sim \mathrm{Gumbel}(0) \tag{3}$$
$$A_1 = \arg\max_a (g(a) + \mathrm{logits}(a)) \tag{4}$$
$$\vdots$$
$$A_n = \arg\max_{a \notin \{A_1, \ldots, A_{n-1}\}} (g(a) + \mathrm{logits}(a)). \tag{5}$$

We will denote the set of $n$ top actions by $\mathtt{argtop}(g + \mathrm{logits}, n) = \{A_1, A_2, \ldots, A_n\}$.

## 3 PLANNING AT THE ROOT

We are interested in improving AlphaZero Monte-Carlo Tree Search (MCTS). In this section we will focus on the action selection at the root of the search tree.

### 3.1 PROBLEM SETTING

Both AlphaZero and MuZero have access to a policy network. At the root of the search tree, they can explore $n$ simulations, before selecting an action for the real environment. We will formalize the

problem as a deterministic bandit with a predictor and we will later extend it to a stochastic bandit and MCTS.

**Bandit.** A $k$-armed deterministic bandit is a vector of Q-values $q \in \mathbb{R}^k$, such that $q(a)$ is the Q-value of the action $a$. The agent interacts with the bandit in $n$ simulations (aka rounds). In each simulation $t \in \{1, \ldots, n\}$, the agent selects an action $A_t \in \{0, \ldots, k-1\}$ and visits the action to observe the Q-value $q(A_t)$.

**The objective** is to maximize the Q-value from a special last action $A_{n+1}$. That means we want to maximize $\mathbb{E}[q(A_{n+1})]$. This objective is equivalent to minimization of *simple regret*. The simple regret differs from the *cumulative regret* from all $n$ simulations. Bubeck et al. (2011), Hay & Russell (2011), and Tolpin & Shimony (2012) already argued that at the root of the search tree we care about the simple regret.

The problem becomes interesting when the number of possible actions is larger than the number of simulations, i.e., when $k > n$. For example, 19x19 Go has 362 possible actions and we will do experiments with as few as $n = 2$ simulations. Fortunately, the policy network can help.

**Predictor.** In the *bandit-with-predictor* setting (Rosin, 2011), the agent is equipped with a predictor: the policy network. Before any interaction with the bandit, the policy network predicts the best action by producing a probability distribution $\pi$. The agent can use the policy network predictions to make more informed decisions.

**Policy improvement.** Naturally, we would like to have an agent that acts better than, or as well as, the policy network. We would like to obtain a policy improvement. If the agent's action selection produces a *policy improvement*, then

$$\mathbb{E}\left[q(A_{n+1})\right] \geq \sum_a \pi(a)q(a), \tag{6}$$

where the probability $\pi(a)$ is the policy network prediction for the action $a$.[1] The policy network can then keep improving by modeling an improved policy.

### 3.2 Motivating Counterexample

We will show that the commonly used heuristics fail to produce a policy improvement.

**Example 1.** Acting with the best action from the top-$n$ most probable actions fails to produce a policy improvement. Let's demonstrate that. Let $q = (0, 0, 1)$ be the Q-values and let $\pi = (0.5, 0.3, 0.2)$ be the probabilities produced by the policy network. The value of the policy network is $\sum_a \pi(a)q(a) = 0.2$. For $n = 2$ simulations, the set of the most probable actions is $\{0, 1\}$. With that, the heuristic would select $A_{n+1} = \arg\max_{a \in \{0,1\}} q(a)$. The expected value of such action is $\mathbb{E}\left[q(A_{n+1})\right] = 0$, which is worse than the value of the policy network.

You can find other counterexamples by generating random $q$ and $\pi$ vectors and testing the policy improvement (Inequality 6). The AlphaZero action selection is explained in Appendix A.

### 3.3 Planning with Gumbel

We will design a policy improvement algorithm for the deterministic bandit with a predictor $\pi$. After $n$ simulations, the algorithm should propose an action $A_{n+1}$ with $\mathbb{E}\left[q(A_{n+1})\right] \geq \sum_a \pi(a)q(a)$.

One possibility is to sample $n$ actions from $\pi$, and then to select from the sampled actions the action with the highest $q(a)$. Instead of sampling with replacement, we can reduce the variance by sampling without replacement.

Still, the sampled actions contain a limited amount of information about $\pi$. We should exploit the knowledge of $\pi$ and its logits when selecting $A_{n+1}$. The main idea is to sample $n$ actions without replacement by using the Gumbel-Top-k trick, and then to use the *same* Gumbel $g$ to select the action with the highest $g(a) + \text{logits}(a) + \sigma(q(a))$. The $\sigma$ can be any monotonically increasing transformation. The pseudocode for the algorithm is in Algorithm 1.

---

[1] Inequality 6 can be strict, if we assume that an action has a positive advantage and its $\pi(a) > 0$.

---

**Algorithm 1** Policy Improvement by Planning with Gumbel

---

**Require:** $k$: number of actions.
**Require:** $n \leq k$: number of simulations.
**Require:** logits $\in \mathbb{R}^k$: predictor logits from a policy network $\pi$.
  Sample $k$ Gumbel variables:
    $(g \in \mathbb{R}^k) \sim \text{Gumbel}(0)$
  Find $n$ actions with the highest $g(a) + \text{logits}(a)$:
    $\mathcal{A}_{\text{topn}} = \texttt{argtop}(g + \text{logits}, n)$
  Get $q(a)$ for each $a \in \mathcal{A}_{\text{topn}}$ by visiting the actions.
  From the $\mathcal{A}_{\text{topn}}$ actions, find the action with the highest $g(a) + \text{logits}(a) + \sigma(q(a))$:
    $A_{n+1} = \arg\max_{a \in \mathcal{A}_{\text{topn}}}(g(a) + \text{logits}(a) + \sigma(q(a)))$
  **return** $A_{n+1}$

---

The algorithm produces a policy improvement, because

$$q(\arg\max_{a \in \mathcal{A}_{\text{topn}}}(g(a) + \text{logits}(a) + \sigma(q(a)))) \geq q(\arg\max_{a \in \mathcal{A}\text{topn}}(g(a) + \text{logits}(a))). \tag{7}$$

This holds for any Gumbel $g$, so it holds also for expectations: $\mathbb{E}[q(A_{n+1})] \geq \mathbb{E}_{A \sim \pi}[q(A)]$. The $\arg\max_{a \in \mathcal{A}\text{topn}}(g(a) + \text{logits}(a))$ is equivalent to sampling from the policy network $\pi$ (see the Gumbel-Max trick or Appendix B). By using the same Gumbel vector $g$ in the $\texttt{argtop}$ and $\arg\max$, we avoid a double-counting bias.

The prior knowledge contained in the logits can help on partially observable environments, or when working with approximate or stochastic Q-values.

### 3.4 PLANNING ON A STOCHASTIC BANDIT

We can now extend the algorithm to a stochastic bandit. A stochastic bandit provides only a stochastic estimate of the expected Q-value $q(a)$. In that setting, we will use the empirical mean $\hat{q}(a)$ instead of $q(a)$. Obviously, the empirical mean would be better estimated, if visiting an action multiple times. We have to choose which actions to visit and how many times. We can control this in two places. First, we can control the number of actions sampled without replacement. Second, we can use a bandit algorithm to efficiently explore the set of sampled actions.

There are multiple bandit algorithms for simple regret minimization. In our preliminary experiments, Sequential Halving (Karnin et al., 2013) was easier to tune than UCB-E (Audibert et al., 2010) and UCB$\sqrt{\cdot}$ (Tolpin & Shimony, 2012). Conveniently, Sequential Halving does not have problem-dependent hyperparameters.

We present Sequential Halving with Gumbel in Algorithm 2, with an illustration in Figure 1. Sequential Halving is used to identify the action with the highest $g(a) + \text{logits}(a) + \sigma(\hat{q}(a))$.

---

**Algorithm 2** Sequential Halving with Gumbel

---

**Require:** $k$: number of actions.
**Require:** $m \leq k$: number of actions sampled without replacement.
**Require:** $n$: number of simulations.
**Require:** logits $\in \mathbb{R}^k$: predictor logits from a policy network $\pi$.
  Sample $k$ Gumbel variables:
    $(g \in \mathbb{R}^k) \sim \text{Gumbel}(0)$
  Find $m$ actions with the highest $g(a) + \text{logits}(a)$:
    $\mathcal{A}_{\text{topm}} = \texttt{argtop}(g + \text{logits}, m)$
  Use Sequential Halving with $n$ simulations to identify the best action from the $\mathcal{A}_{\text{topm}}$ actions,
  by comparing $g(a) + \text{logits}(a) + \sigma(\hat{q}(a))$.
  $A_{n+1} = \arg\max_{a \in Remaining}(g(a) + \text{logits}(a) + \sigma(\hat{q}(a)))$
  **return** $A_{n+1}$

---

For a concrete instantiation of $\sigma$, we use

$$\sigma(\hat{q}(a)) = (c_{\text{visit}} + \max_b N(b))c_{\text{scale}}\hat{q}(a), \tag{8}$$

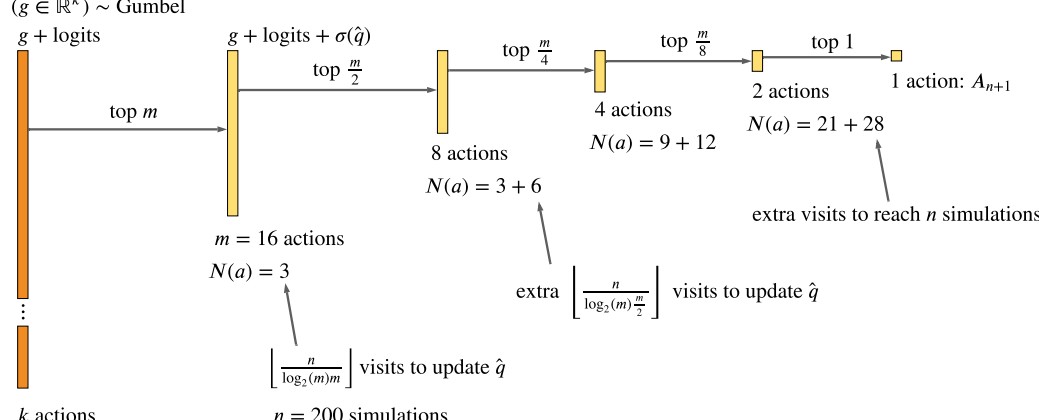

Figure 1: The number of considered actions and their visit counts $N(a)$, when using Sequential Halving with Gumbel on a $k$-armed stochastic bandit. The search uses $n = 200$ simulations and starts by sampling $m = 16$ actions without replacement. Sequential Halving divides the budget of $n$ simulations equally to $\log_2(m)$ phases. In each phase, all considered actions are visited equally often. After each phase, one half of the actions is rejected. From the original $k$ actions, only the best actions will remain.

where $\max_b N(b)$ is the visit count of the most visited action. The transformation progressively increases the scale for $\hat{q}(a)$ and reduces the effect of the prior policy. This scale is inspired by the policy updates in MPO (Abdolmaleki et al., 2018; Vieillard et al., 2020). The finite scale for the Q-values provides regularized policy optimization and puts into effect the prior knowledge contained in the logits. Experimentally, $c_{\text{visit}} = 50$, $c_{\text{scale}} = 1.0$ allowed us to use the same hyperparameters even if changing the number of simulations.

## 4 LEARNING AN IMPROVED POLICY

After the search, we have $A_{n+1}$ from a (potentially) improved policy. Like AlphaZero, we would like to distill the improved policy to the policy network. One possibility is to train the policy network $\pi$ to predict the $A_{n+1}$. That defines a simple policy loss:

$$L_{\text{simple}}(\pi) = -\log \pi(A_{n+1}).$$ (9)

**Using completed Q-values.** We will explain a different way to train the policy network, by extracting more knowledge from the search. Beside $A_{n+1}$, the search also gives us $q(a)$ (or its approximation) for the visited actions. We can construct an improved policy by first completing the vector of Q-values:

$$\text{completedQ}(a) = \begin{cases} q(a) & \text{if } N(a) > 0 \\ v_\pi, & \text{otherwise,} \end{cases}$$ (10)

where the unknown Q-values of the unvisited actions are replaced by $v_\pi = \sum_a \pi(a)q(a)$. While in practice we do not have the exact $v_\pi$, we have instead its approximation $\hat{v}_\pi$ from a value network. Even when training on off-policy data we devised a helpful $v_\pi$ approximation (Appendix D).

With the completed Q-values, a new improved policy is constructed by

$$\pi' = \text{softmax}(\text{logits} + \sigma(\text{completedQ})),$$ (11)

where $\sigma$ is a monotonically increasing transformation. We provide a proof of a policy improvement in Appendix C. Intuitively, the completed Q-values give zero advantage to the unvisited actions.

After constructing the new improved policy $\pi'$, we can distill it to the policy network $\pi$:

$$L_{\text{completed}}(\pi) = \text{KL}(\pi', \pi).$$ (12)

This loss trains all actions, not only the action $A_{n+1}$. Later, we will investigate the effect of the loss in Figure 3a.

## 5   PLANNING AT NON-ROOT NODES

To design an action selection for the non-root nodes of a search tree, we take inspiration from Grill et al. (2020). That allows us to interpret MCTS as regularized policy optimization. At a non-root node, we construct an improved policy $\pi'$ by using the completed Q-values (Equation 11).

To select an action at the non-root node, one possibility is to sample the action from $\pi'$. However, sampling at non-root nodes adds unwanted variance to the estimated Q-values. Instead, we can design a deterministic action selection with the smallest mean-squared-error between the $\pi'$ probabilities and the produced normalized visit counts. Such action selection would select

$$\arg\min_a \sum_b \left( \pi'(b) - \underbrace{\frac{N(b) + \mathbb{I}\{a = b\}}{1 + \sum_c N(c)}}_{\text{Normalized visit counts, if taking } a.} \right)^2, \tag{13}$$

where the indicator $\mathbb{I}\{a = b\}$ is 1 if $a = b$, and zero otherwise. After a bit of algebra (Appendix E), we obtain a simpler, more efficient expression:

$$\arg\max_a \left( \pi'(a) - \frac{N(a)}{1 + \sum_b N(b)} \right). \tag{14}$$

This deterministic action selection selects the actions proportionally to $\pi'$ and avoids an extra variance. We recommend a deterministic action selection only for non-root nodes. At the root node, the Gumbel noise is helpful for trying different actions in different episodes, while ensuring an improved expected value.

## 6   RELATED WORK

Rosin (2011) introduced the bandit with a predictor and designed PUCB ("Predictor + UCB") for cumulative regret minimization. AlphaGo (Silver et al., 2016), AlphaGo Zero (Silver et al., 2017), AlphaZero (Silver et al., 2018), and MuZero (Schrittwieser et al., 2020) used a deep policy network as the predictor in a variant of the PUCB algorithm. Bertsekas (2019; 2021; 2022) provides an in-depth discussion of policy iteration, policy improvement, and their connection to rollout.

If not using a predictor, UCT (Kocsis & Szepesvári, 2006) would need to visit each action before being able to compare them. Rapid Action Value Estimation (Gelly & Silver, 2011) then helps to form rough estimates of the action values by aggregating statistics from all future states. Gelly & Silver (2011) also initialized the action value estimates with a heuristic evaluation function. The best heuristic used a learned linear network. Hamrick et al. (2020) later extended it to a deep Q-network.

Cazenave (2014) and Pepels et al. (2014) applied Sequential Halving to MCTS. Fabiano & Cazenave (2021) introduced Sequential Halving Using Scores. The 'scores' can be any prior offset to the Q-values. The way to obtain scores from a policy network was left as an open problem. We can now view the $g + \text{logits}$ as special scores.

MCTS is related to regularized policy optimization. Grill et al. (2020) analyzed AlphaZero tree search and discovered that AlphaZero approximates a regularized policy optimization. The approximation error is large if using a small number of simulations. To avoid the approximation error, Grill et al. (2020) used a regularized policy optimization directly inside the tree search. In the setting without a predictor, Xiao et al. (2019) compared UCT to a new MCTS with an entropy regularizer. Dam et al. (2021) generalized it to relative entropy and Tsallis entropy. Regularized policy optimization is helpful when working with approximate Q-values (Vieillard et al., 2020) or when doing an approximate policy iteration (Kakade & Langford, 2002; Schulman et al., 2015).

TreeQN (Farquhar et al., 2018) uses a breadth-first search inside a network architecture. The network can do a lot of computation before producing a Q-value. To reduce the computation demands, Dynamic Planning Networks (Tasfi & Capretz, 2018) extended TreeQN to sample only some actions. To approximate the gradient, Dynamic Planning Networks use Gumbel-Softmax (Maddison et al., 2017; Jang et al., 2017). Although we use Gumbel variables, we do not employ approximate gradients from Gumbel-Softmax. We use the Gumbel-Top-k trick to construct efficient planning with a provable policy improvement.

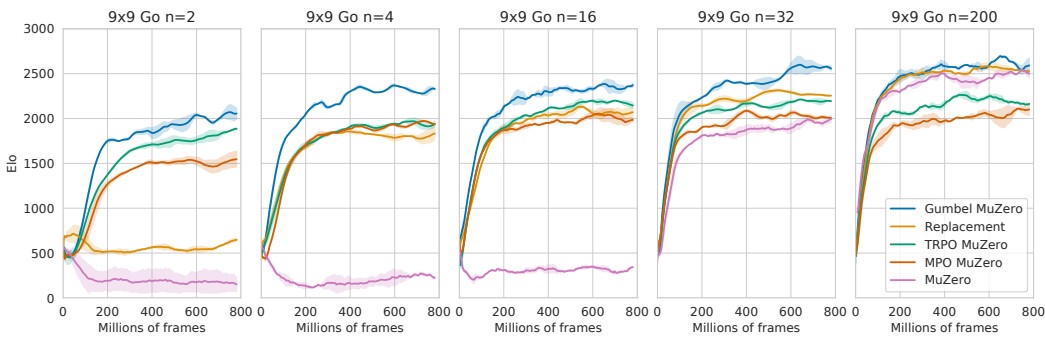

Figure 2: Elo on 9x9 Go, when training with $n \in \{2, 4, 16, 32, 200\}$ simulations. Evaluation uses 800 simulations. Shades denote standard errors from 2 seeds.

For sampling without replacement, the unordered set estimator by Kool et al. (2020) provides an elegant, unbiased estimate of a gradient. However, gradient descent needs multiple steps to reach the solution of a regularized policy optimization problem (Tomar et al., 2020). Furthermore, the exact computation of the unordered set estimator requires $O(2^m)$ operations, which would be prohibitively expensive for $m = 16$. Practical time complexity can be achieved by using an importance-weighted estimator (Vieira, 2017; Nauman & Den Hengst, 2020).

**Continuous actions** can be supported by sampling $k$ actions with replacement and then using the sampled actions as discrete actions with uniform logits for the rest of the search. This was done in Sampled MuZero (Hubert et al., 2021). Similarly, Critic Weighted Policy (Wang et al., 2020) uses sampling with replacement.

## 7 EXPERIMENTS

In the experiments, we compare AlphaZero or MuZero to the proposed planning with Gumbel and other alternatives:

**MuZero:** The newest version of MuZero (Schrittwieser et al., 2021), with ResNet v2 style pre-activation residual blocks (He et al., 2016) and the Adam optimizer (Kingma & Ba, 2014). **Gumbel MuZero:** MuZero with the modified root of the search tree to use Sequential Halving with Gumbel. The policy loss uses the completed Q-values (Equation 12). **Gumbel MuZero sampled with replacement (Replacement):** An ablation to Gumbel MuZero by sampling $m$ actions with replacement, as in Sampled MuZero (Hubert et al., 2021). **TRPO MuZero:** MuZero with modified learning, acting, and the root of the search tree to use the regularized policy optimization with the TRPO regularizer $\text{KL}(\pi, \pi_{\text{new}})$ (Schulman et al., 2015; Grill et al., 2020). **MPO MuZero:** TRPO MuZero but with the MPO regularizer $\text{KL}(\pi_{\text{new}}, \pi)$ (Abdolmaleki et al., 2018; Grill et al., 2020). **Full Gumbel MuZero:** Gumbel MuZero with a principled action selection also for the non-root search nodes (Section 5). In the plots, we will show Full Gumbel MuZero only if it produces results significantly different from Gumbel MuZero.

We conducted the experiments on Go, chess, and Atari. We present the main results here and we report additional ablations and experimental details in Appendix F.

### 7.1 9x9 GO

On Go, we use Elo to compare MuZero and other agents. While an agent trains by self-play, its Elo is computed by evaluation versus reference opponents. One of the opponents is Pachi (Baudiš & Gailly, 2011) with 10k simulations per move. We anchored the Elo to have this Pachi at 1000 Elo. For example, a difference of 500 Elo corresponds to a 95% win probability for the player with the higher Elo.[2]

---

[2]The corresponding win probability is $\frac{1}{1+10^{-\frac{EloDifference}{400}}}$ (Elo, 1978).

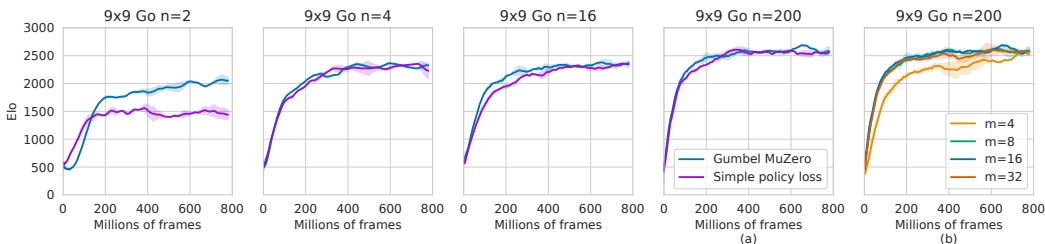

Figure 3: Gumbel MuZero ablations on 9x9 Go. **(a)** Policy loss ablations, when training with $n \in \{2, 4, 16, 200\}$ simulations. Gumbel MuZero uses the policy loss with completed Q-values. **(b)** Sensitivity to the number of sampled actions. Gumbel MuZero samples $m$ actions without replacement.

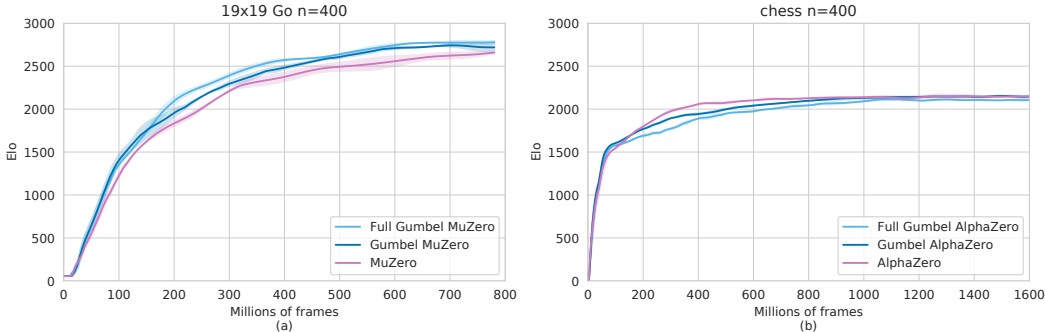

Figure 4: Large-scale experiments with $n = 400$ simulations per move. **(a)** Elo on 19x19 Go, when training MuZero. **(b)** Elo on chess, when training AlphaZero.

In Figure 2 we investigate the impact of the number of simulations on the obtained Elo. When training an agent by self-play, the agent uses $n$ simulations per move. In the five plots, the $n$ varies from 2 to 200. In evaluation, we allow all agents to use 800 simulations. The speed of the evaluation does not affect the speed of training. In the 9x9 Go results, MuZero fails to learn from 16 or less simulations. Strikingly, Gumbel MuZero learns reliably even with 2 simulations.

In Figure 3a we compare the simple policy loss (Equation 9) and the policy loss with the completed Q-values (Equation 12). The simple policy loss would be enough for many applications. We used the completed Q-values also in TRPO MuZero and MPO MuZero. Without the completed Q-values, TRPO MuZero and MPO MuZero would fail to produce a policy improvement.

In Figure 3b we study Gumbel MuZero's sensitivity to the number of sampled actions. When sampling $m = 4$ actions without replacement, the simulation budget is spent on the small number of actions. The learning was then slower. In all other Go experiments, we sample $m = \min(n, 16)$ actions without replacement.

## 7.2 LARGE-SCALE 19x19 GO AND CHESS

In Figure 4a we demonstrate that Gumbel MuZero is not worse than MuZero on 19x19 Go. MuZero is excellent on 19x19 Go and Gumbel MuZero reaches or exceeds its performance. The Elo is still anchored to have Pachi at 1000 Elo.

Similarly, in Figure 4b we show Gumbel AlphaZero performance on chess. We train AlphaZero on chess, because AlphaZero learns faster than MuZero on chess. On Go, MuZero learns faster than AlphaZero.

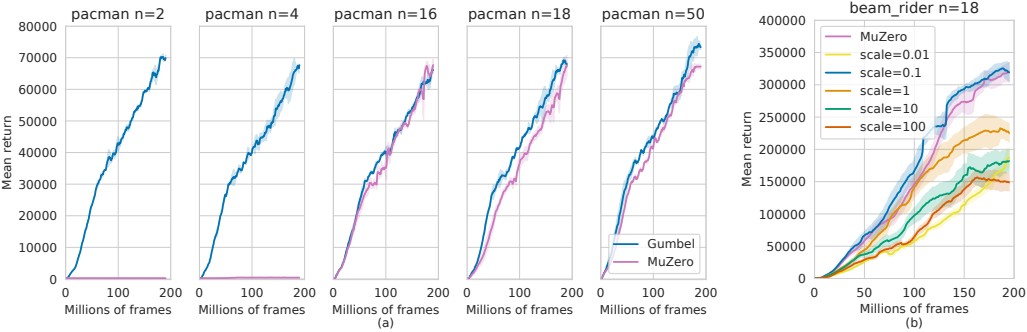

Figure 5: Atari results. **(a)** Mean return on `ms_pacman`, when training Gumbel MuZero and MuZero with $n \in \{2, 4, 16, 18, 50\}$ simulations. MuZero fails to learn from 4 or less simulations. **(b)** Mean return on `beam_rider` for Gumbel MuZero with $c_{scale} \in \{0.01, 0.1, 1, 10, 100\}$, compared to MuZero with $n = 50$ simulations. Shades denote standard errors from 10 seeds.

### 7.3 ATARI

Our last set of experiments is on Atari. We use the Arcade Learning Environment (Bellemare et al., 2013) with sticky actions (Machado et al., 2018). The network sizes and hyperparameters match the MuZero setup by Schrittwieser et al. (2021). On Atari, MuZero does not use more simulations at evaluation. The reported score is the mean return from the last 200 training episodes. MuZero with $n = 50$ simulations works well on Atari and establishes the state of the art (Schrittwieser et al., 2021).

In Figure 5a we show the obtained mean return on `ms_pacman`. Gumbel MuZero again learns reliably even with $n = 2$ simulations. Atari has only 18 actions, so we sample $m = \min(n, 18)$ actions without replacement. In the experiments with $n \leq 18$, Gumbel MuZero selects $A_{n+1}$ from the $n$ visited actions, without any Sequential Halving. This confirms that planning with Gumbel is the key ingredient responsible for the policy improvement from a small number of simulations.

Atari is challenging, because different games can have very different reward scales. MuZero normalizes the Q-values by dividing them by $\max(\hat{v}_\pi, \max_a \hat{q}(a)) - \min(\hat{v}_\pi, \min_a \hat{q}(a))$ found inside the tree search (Schrittwieser et al., 2020). A normalized advantage is then in $[-1, 1]$. For Gumbel MuZero, we use the same normalization and we scale the normalized Q-values by $c_{visit} = 50$ and $c_{scale} = 0.1$. A scaled normalized advantage is then approximately in $[-5, 5]$. Thanks to the bounded advantage, Gumbel MuZero has a bounded total variation distance between $\pi$ and $\pi'$ (Hessel et al., 2021).

In Figure 5b we use `beam_rider` as an example of a partially observable game and we study the importance of the prior knowledge contained in the logits. Gumbel MuZero selects an action based on $g(a) + \text{logits}(a) + (c_{visit} + \max_b N(b)) c_{scale} \hat{q}(a)$ (Equation 8). If $c_{scale}$ is large, Gumbel MuZero focuses on $\hat{q}(a)$ and neglects the logits. Indeed, Gumbel MuZero performance is worse on `beam_rider` if using large $c_{scale}$. In the future, we can try normalizing the Q-values by the standard deviation of an advantage estimator and we can try clipping the normalized advantages, as in Muesli (Hessel et al., 2021).

## 8 CONCLUSION

We redesigned AlphaZero tree search. With the principle of policy improvement, we replaced five heuristic mechanisms in AlphaZero. On Go, chess, and Atari, we validated that Gumbel MuZero and Gumbel AlphaZero keep improving, even when learning from two simulations. On top of that, Gumbel MuZero provides a principled way to achieve state-of-the-art results. We hope that future research will benefit from the clean theoretical foundation, the faster experimentation with a small number of simulations, and the released open-source code.[3]

---

[3]`https://github.com/deepmind/mctx`

ACKNOWLEDGMENTS

We would like to thank Fabio Viola, Matteo Hessel, and Laurent Sifre for building the core of the open-sourced Monte Carlo tree search. Loic Matthey and Benjamin Van Roy provided helpful comments on the paper draft. Thomas Hubert, Iurii Kemaev, Tor Lattimore, and Jean-bastien Grill shared their wisdom.

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

APPENDIX

**Content**

- A - AlphaZero action selection
- B - Policy improvement proof for planning with Gumbel
- C - Policy improvement proof for completed Q-values
- D - Mixed value approximation
- E - Derivation of the deterministic action selection
- F - Experimental details

## A  ALPHAZERO ACTION SELECTION

We will explain AlphaZero's (Silver et al., 2018) action selection here. On the deterministic bandit, AlphaZero would select actions $A_1, \ldots, A_n$ by

$$\arg\max_a \left[ q^?(a) + c_1 \pi(a) \frac{\sqrt{1 + \sum_b N(b)}}{1 + N(a)} \right], \tag{15}$$

where $q^?(a) \in [0, 1]$ is $q(a)$ for the already visited actions and zero otherwise. The $c_1 > 0$ is a factor independent of $a$. After the search, AlphaZero acts by sampling from an (annealed) categorical distribution based upon the visit counts of the root actions.

At the root of the search tree, AlphaZero perturbs $\pi$ by Dirichlet noise to avoid visiting always the most probable actions. That does not ensure a policy improvement, because AlphaZero with Dirichlet noise can produce a worse policy by adding noise to a potentially optimal policy network.

## B  POLICY IMPROVEMENT PROOF FOR PLANNING WITH GUMBEL

We will prove that Algorithm 1 generates $A_{n+1}$ such that $\mathbb{E}[q(A_{n+1})] \geq \mathbb{E}_{A \sim \pi}[q(A)]$. For the right-hand side, the Gumbel-Max trick tells us that $\mathbb{E}_{A \sim \pi}[q(A)]$ is equal to $\mathbb{E}_{(g \in \mathbb{R}^k) \sim \text{Gumbel}(0)}[q(\arg\max_a(g(a) + \text{logits}(a))]$. First, we will show that we can replace the $\arg\max_a$ with $\arg\max_{a \in \mathcal{A}_{\text{topn}}}$. Remember that $\mathcal{A}_{\text{topn}}$ is defined as $\mathcal{A}_{\text{topn}} = \texttt{argtop}(g + \text{logits}, n)$ and that we use the same Gumbel vector $g$ in the $\texttt{argtop}$ and $\arg\max$. The set $\mathcal{A}_{\text{topn}}$ then includes the action with the highest $g(a) + \text{logits}(a)$ and we can replace the $\arg\max_a$ with $\arg\max_{a \in \mathcal{A}_{\text{topn}}}$.

After these rewrites, we have to prove that

$$\mathbb{E}[q(A_{n+1})] \geq \mathbb{E}_{(g \in \mathbb{R}^k) \sim \text{Gumbel}(0)}[q(\arg\max_{a \in \mathcal{A}_{\text{topn}}}(g(a) + \text{logits}(a))]. \tag{16}$$

On the left-hand side, $\mathbb{E}[q(A_{n+1})]$ is equal to

$$\mathbb{E}_{(g \in \mathbb{R}^k) \sim \text{Gumbel}(0)}[q(\arg\max_{a \in \mathcal{A}_{\text{topn}}}(g(a) + \text{logits}(a) + \sigma(q(a))))]. \tag{17}$$

We can finish the proof by proving that for any vector $g \in \mathbb{R}^k$ we have

$$q(\arg\max_{a \in \mathcal{A}_{\text{topn}}}(g(a) + \text{logits}(a) + \sigma(q(a)))) \geq q(\arg\max_{a \in \mathcal{A}_{\text{topn}}}(g(a) + \text{logits}(a))). \tag{18}$$

This is true, because $\sigma$ is a monotonically increasing transformation.

## C  POLICY IMPROVEMENT PROOF FOR COMPLETED Q-VALUES

We will prove that $\pi'_{\text{completed}} = \text{softmax}(\text{logits} + \sigma(\text{completedQ}))$ produces a policy improvement. We will start by showing that $\pi'_{\text{completed}}$ is produced by a specific instance of Algorithm 3. We will then prove that any instance of Algorithm 3 produces a policy improvement.

---

**Algorithm 3** Policy improvement when having $v_\pi$

---

**Require:** $\pi, v_\pi$.
**Require:** $q(a)$ for each visited action. The visited actions can be from any distribution.
  Initialize $\pi'$ with $\pi$.
  For the visited actions:
    If $q(a) > v_\pi$, increase the $\pi'(a)$ logit.
    If $q(a) < v_\pi$, decrease the $\pi'(a)$ logit.
  **return** $\pi'$

---

### C.1 SPECIFIC INSTANCE

Algorithm 3 is more general than the usage of the completed Q-values. Specifically, Algorithm 3 would produce $\pi'_{\text{completed}}$, if updating the logits by $\sigma(\text{completedQ}(a)) - \sigma(v_\pi)$. This update increases the logit, if $q(a) > v_\pi$. This update decreases the logit, if $q(a) < v_\pi$. And the update does not modify the logits of the unvisited actions. The resulting $\text{softmax}(\text{logits} + \sigma(\text{completedQ}) - \sigma(v_\pi))$ is equal to $\text{softmax}(\text{logits} + \sigma(\text{completedQ}))$, because the constant offset $\sigma(v_\pi)$ does not change the softmax output.

### C.2 POLICY IMPROVEMENT PROOF FOR ANY INSTANCE

We will now prove that $\pi'$ from Algorithm 3 satisfies

$$\sum_a \pi'(a)q(a) \geq \sum_a \pi(a)q(a). \tag{19}$$

Notice that $\pi'(a)$ for any unvisited action $a$ will be $c_z\pi(a)$, with a normalization constant $c_z > 0$.

**For one visited action:** Let's denote the visited (aka expanded) action by $a_{\text{ex}}$. First, if $\pi(a_{\text{ex}}) = 1$ then $v_\pi = q(a_{\text{ex}})$ and the policy will remain unchanged.

Let's now consider the case with $\pi(a_{\text{ex}}) < 1$. The $v_\pi$ can be rewritten as

$$v_\pi = \pi(a_{\text{ex}})q(a_{\text{ex}}) + (1 - \pi(a_{\text{ex}})) \sum_{a \neq a_{\text{ex}}} \frac{\pi(a)q(a)}{\sum_{b \neq a_{\text{ex}}} \pi(b)}. \tag{20}$$

Let's denote the weighted sum by $q_{\text{miss}}$:

$$q_{\text{miss}} = \sum_{a \neq a_{\text{ex}}} \frac{\pi(a)q(a)}{\sum_{b \neq a_{\text{ex}}} \pi(b)}. \tag{21}$$

We notice that the $q_{\text{miss}}$ does not change if scaling $\pi$ by a constant $c_z > 0$.

We will now rewrite the left-hand side of Inequality 19 to use $q_{\text{miss}}$:

$$\sum_a \pi'(a)q(a) = \tag{22}$$

$$= \pi'(a_{\text{ex}})q(a_{\text{ex}}) + (1 - \pi'(a_{\text{ex}})) \sum_{a \neq a_{\text{ex}}} \frac{c_z\pi(a)q(a)}{\sum_{b \neq a_{\text{ex}}} c_z\pi(b)} \tag{23}$$

$$= \pi'(a_{\text{ex}})q(a_{\text{ex}}) + (1 - \pi'(a_{\text{ex}}))q_{\text{miss}} \tag{24}$$

$$= \pi'(a_{\text{ex}})(q(a_{\text{ex}}) - q_{\text{miss}}) + q_{\text{miss}}. \tag{25}$$

The right-hand side of Inequality 19 can be also rewritten:

$$v_\pi = \pi(a_{\text{ex}})q(a_{\text{ex}}) + (1 - \pi(a_{\text{ex}}))q_{\text{miss}} \tag{26}$$

$$= \pi(a_{\text{ex}})(q(a_{\text{ex}}) - q_{\text{miss}}) + q_{\text{miss}}. \tag{27}$$

With these rewrites, Inequality 19 becomes

$$\pi'(a_{\text{ex}})(q(a_{\text{ex}}) - q_{\text{miss}}) \geq \pi(a_{\text{ex}})(q(a_{\text{ex}}) - q_{\text{miss}}). \tag{28}$$

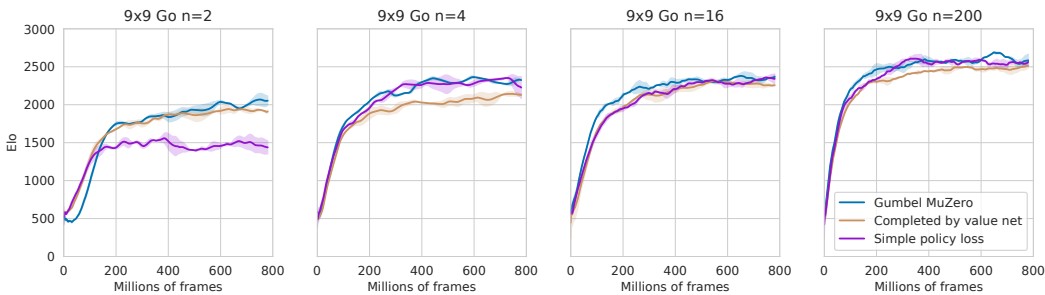

Figure 6: Detailed policy loss ablations. Gumbel MuZero uses the policy loss with Q-values completed by the $v_{\mathrm{mix}}$ value estimator from Appendix D. That works better than Q-values completed by the raw value network $\hat{v}_\pi$.

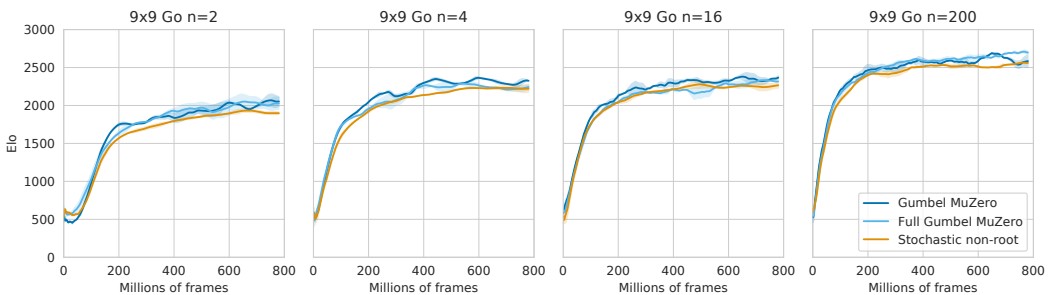

Figure 7: A comparison of different action selections at the non-root nodes. Gumbel MuZero uses the unmodified (deterministic) MuZero action selection at non-root nodes. Full Gumbel MuZero uses the deterministic action selection from Equation 14, which we compare to stochastic sampling from $\pi'$ at non-root nodes.

The $q(a_{\mathrm{ex}}) - q_{\mathrm{miss}}$ can negative, zero or positive.
If $q(a_{\mathrm{ex}}) = q_{\mathrm{miss}}$, the inequality is satisfied.
If $q(a_{\mathrm{ex}}) > q_{\mathrm{miss}}$, we want $\pi'(a_{\mathrm{ex}}) \geq \pi(a_{\mathrm{ex}})$.
If $q(a_{\mathrm{ex}}) < q_{\mathrm{miss}}$, we want $\pi'(a_{\mathrm{ex}}) \leq \pi(a_{\mathrm{ex}})$.

We do not know $q_{\mathrm{miss}}$ so we cannot use it in an algorithm. We will instead show that $q(a_{\mathrm{ex}}) > q_{\mathrm{miss}}$ is equivalent to $q(a_{\mathrm{ex}}) > v_\pi$, when $\pi(a_{\mathrm{ex}}) < 1$:

$$q(a_{\mathrm{ex}}) > v_\pi \tag{29}$$

$$q(a_{\mathrm{ex}}) > \pi(a_{\mathrm{ex}})q(a_{\mathrm{ex}}) + (1 - \pi(a_{\mathrm{ex}}))q_{\mathrm{miss}} \tag{30}$$

$$(1 - \pi(a_{\mathrm{ex}}))q(a_{\mathrm{ex}}) > (1 - \pi(a_{\mathrm{ex}}))q_{\mathrm{miss}} \tag{31}$$

$$q(a_{\mathrm{ex}}) > q_{\mathrm{miss}}. \tag{32}$$

So we directly arrived at Algorithm 3.

**For multiple visited actions:** We will focus on one of the visited actions. If the logits of the other visited actions are unmodified, the algorithm is equivalent to using only one visited action. If the logits of the other visited actions are modified by Algorithm 3, they can further help to improve the policy.

## D    MIXED VALUE APPROXIMATION

We will construct an approximation of $v_\pi$. The exact $v_\pi$ is defined by $v_\pi = \sum_a \pi(a)q(a)$. We have an approximate $\hat{v}_\pi$ from a value network, we know $\pi$, and we have $q(a)$ for the visited actions. With

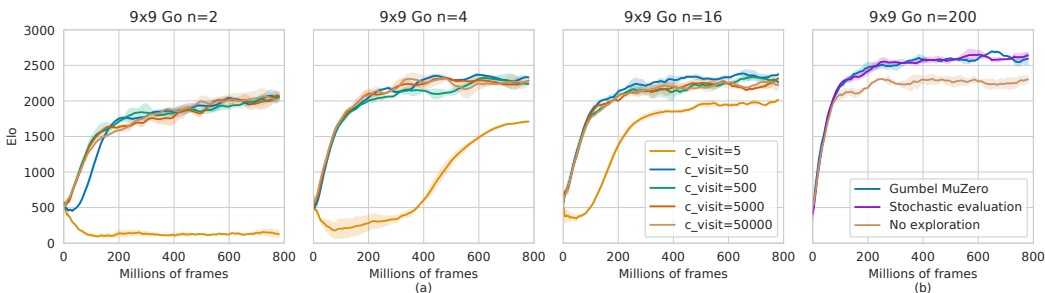

Figure 8: Additional Gumbel MuZero ablations on 9x9 Go. **(a)** Sensitivity to Q-value scaling by $c_{\text{visit}}$. **(b)** On the perfect-information game, Gumbel MuZero used zero Gumbel noise at evaluation. Although, evaluation with stochastic Gumbel noise is not worse. During training, MuZero and Gumbel MuZero benefit from explorative acting proportional to the visit counts.

these inputs, we approximate $v_\pi$ by a consistent estimator:

$$v_{\text{mix}} = \frac{1}{1 + \sum_b N(b)} \left( \hat{v}_\pi + \frac{\sum_b N(b)}{\sum_{b \in \{b : N(b) > 0\}} \pi(b)} \sum_{a \in \{a : N(a) > 0\}} \pi(a) q(a) \right). \tag{33}$$

The estimator interpolates $\hat{v}_\pi$ and the weighted average of the available Q-values. This is an unsophisticated estimator, with results in Figure 6. You are welcome to explore other possibilities.

## E  DERIVATION OF THE DETERMINISTIC ACTION SELECTION

We will derive Equation 14 from Equation 13:

$$\arg\min_a \sum_b \left( \pi'(b) - \frac{N(b) + \mathbb{I}\{a = b\}}{1 + \sum_c N(c)} \right)^2 \tag{34}$$

$$= \arg\min_a \sum_b \left( \left( \pi'(b) - \frac{N(b)}{1 + \sum_c N(c)} \right) - \frac{\mathbb{I}\{a = b\}}{1 + \sum_c N(c)} \right)^2 \tag{35}$$

$$= \arg\min_a \sum_b -2 \left( \pi'(b) - \frac{N(b)}{1 + \sum_c N(c)} \right) \frac{\mathbb{I}\{a = b\}}{1 + \sum_c N(c)} \tag{36}$$

$$= \arg\min_a -\sum_b \left( \pi'(b) - \frac{N(b)}{1 + \sum_c N(c)} \right) \mathbb{I}\{a = b\} \tag{37}$$

$$= \arg\max_a \left( \pi'(a) - \frac{N(a)}{1 + \sum_c N(c)} \right). \tag{38}$$

The simplification was possible, because additive terms independent of $a$ do not affect $\arg\min_a$. While widely applicable, the deterministic action selection provides only a small benefit on 9x9 Go (Figure 7).

## F  EXPERIMENTAL DETAILS

In general, we use hyperparameters consistent with the newest MuZero experiments (Schrittwieser et al., 2021). MuZero's pseudocode is available thanks to Schrittwieser et al. (2020). Gumbel MuZero does not need to set the Dirichlet noise hyperparameters, because Gumbel MuZero does not use Dirichlet noise.

In a tree search, the Q-values $\hat{q}(a)$ are provided by the visited child nodes. We do not modify the structure of AlphaZero's search tree. Inside the tree search, the Q-values are normalized to be from

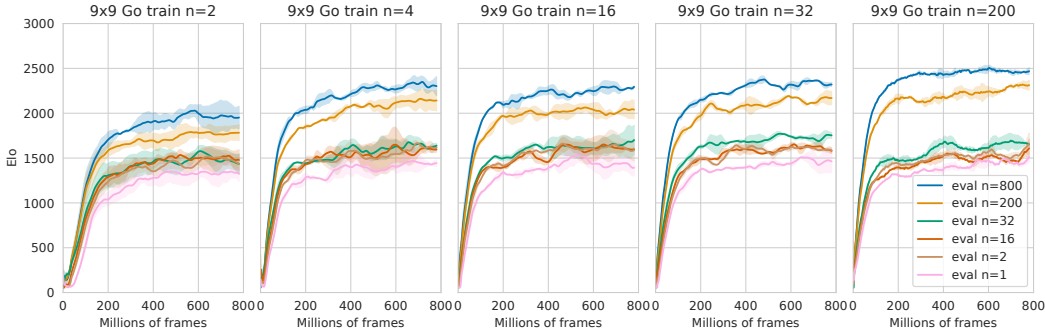

Figure 9: Gumbel MuZero Elo on 9x9 Go, evaluated with different numbers of simulations. The evaluation with $n = 1$ simulation acts with the most probable action from the policy network.

the $[0, 1]$ interval. We use the normalized Q-values also in Full Gumbel MuZero, but the algorithm does not require Q-values from a specific interval. In all Go and chess experiments, Gumbel MuZero scales the Q-values by $c_{\text{visit}} = 50$ and $c_{\text{scale}} = 1.0$. On the perfect-information game of Go, Gumbel MuZero is not very sensitive to the scale of the Q-values. Any $c_{\text{visit}} \geq 50$ produced similar results (Figure 8a).

In each phase of Sequential Halving, we use at least one new visit. For example, in the first phase, we update $\hat{q}(a)$ by $\max\left(1, \left\lfloor \frac{n}{\lceil \log_2(m) \rceil m} \right\rfloor\right)$ visits. This allows us to experiment with an incomplete or no Sequential Halving. When Sequential Halving runs out of the budget of $n$ simulations, we stop the search. The agent then selects as $A_{n+1}$ the action with the highest $g(a) + \text{logits}(a) + \sigma(\hat{q}(a))$ from the set of the most visited actions. Fabiano & Cazenave (2021) provide a different way to deal with the rounding in Sequential Halving.

During training, MuZero acts with explorative actions in the first 30 moves of each self-play game. MuZero samples the explorative actions proportionally to the visit counts, like AlphaGo Zero (Silver et al., 2017). Gumel MuZero benefits from the same exploration (Figure 8b).

Figure 9 shows the importance of the number of simulations at evaluation time. For example, in the first subplot a network is trained with $n = 2$ simulations and the same network is evaluated with 800, 200, 32, 16, 2, and 1 simulations.

To run the experiments, we used Google Cloud Tensor Processing Units v3 (TPUs). On 9x9 Go, MuZero is not limited by lack of data if using 3-times more TPUs for self-play than for training. By using a smaller number of simulations, we can substantially reduce the number of TPUs needed for self-play. Table1 lists the obtained speedups if not being limited by the TPUs for training.

Table 1: The speedup from a smaller number of simulations on 9x9 Go.

|  | Training step speedup |
|---|---|
| MuZero $n = 200$ | 1.0 |
| Full Gumbel MuZero $n = 200$ | 1.0 |
| Gumbel MuZero $n = 200$ | 1.0 |
| Gumbel MuZero $n = 32$ | 5.9 |
| Gumbel MuZero $n = 16$ | 11.3 |
| Gumbel MuZero $n = 8$ | 16.2 |
| Gumbel MuZero $n = 4$ | 24.3 |

## F.1 NETWORK ARCHITECTURE

In all Go and chess experiments, we used a modified version of the AlphaZero network architecture that is roughly 2-times faster at the same accuracy. Concretely, we replace the dense residual blocks

with bottleneck blocks (He et al., 2016), and replace every 8th block with a global broadcasting residual block (Figures 11, 13). The broadcasting block is similar to global pooling or squeeze-and-excitation (Hu et al., 2018), but we found it to be more stable when training from self-play.

Figure 10 provides data to support the architecture change. For each network architecture the figure shows multiple data points, differing by the number of layers. For dense networks, the number of layers is from $\{12, 16, 20, 24, 28, 32, 36, 40, 48\}$. For bottleneck networks, data points with 56 and 64 layers are shown as well.

Both value MSE and policy accuracy improve consistently with larger but slower networks. For any constant number of inferences per second, the bottleneck blocks and the broadcast/pooling blocks lead to significant improvements.

For the architecture search, the Elo was computed at a fixed computational budget of 300ms per move. For example, the budget allows us to do 3200 simulations when using a network with 256 planes and 32 blocks with bottlenecks and broadcasting (achieving $10^4$ inferences/second). Here the Elo is anchored to have the base AlphaZero architecture at 0 Elo; 95% confidence intervals are about 30 Elo wide. We observe that diminishing returns in the prediction accuracy for very large networks lead to decreasing Elo, as the improved predictions fail to compensate for the much lower number of simulations in MCTS. Gumbel AlphaZero was not yet used for the architecture search.

Given the efficient network architecture, we used a small 6 layer network on 9x9 Go, and a bigger 32 layer network in the large-scale 19x19 Go experiments. The networks used 256 hidden planes, 128 bottleneck planes and a broadcasting block in every 8th layer.

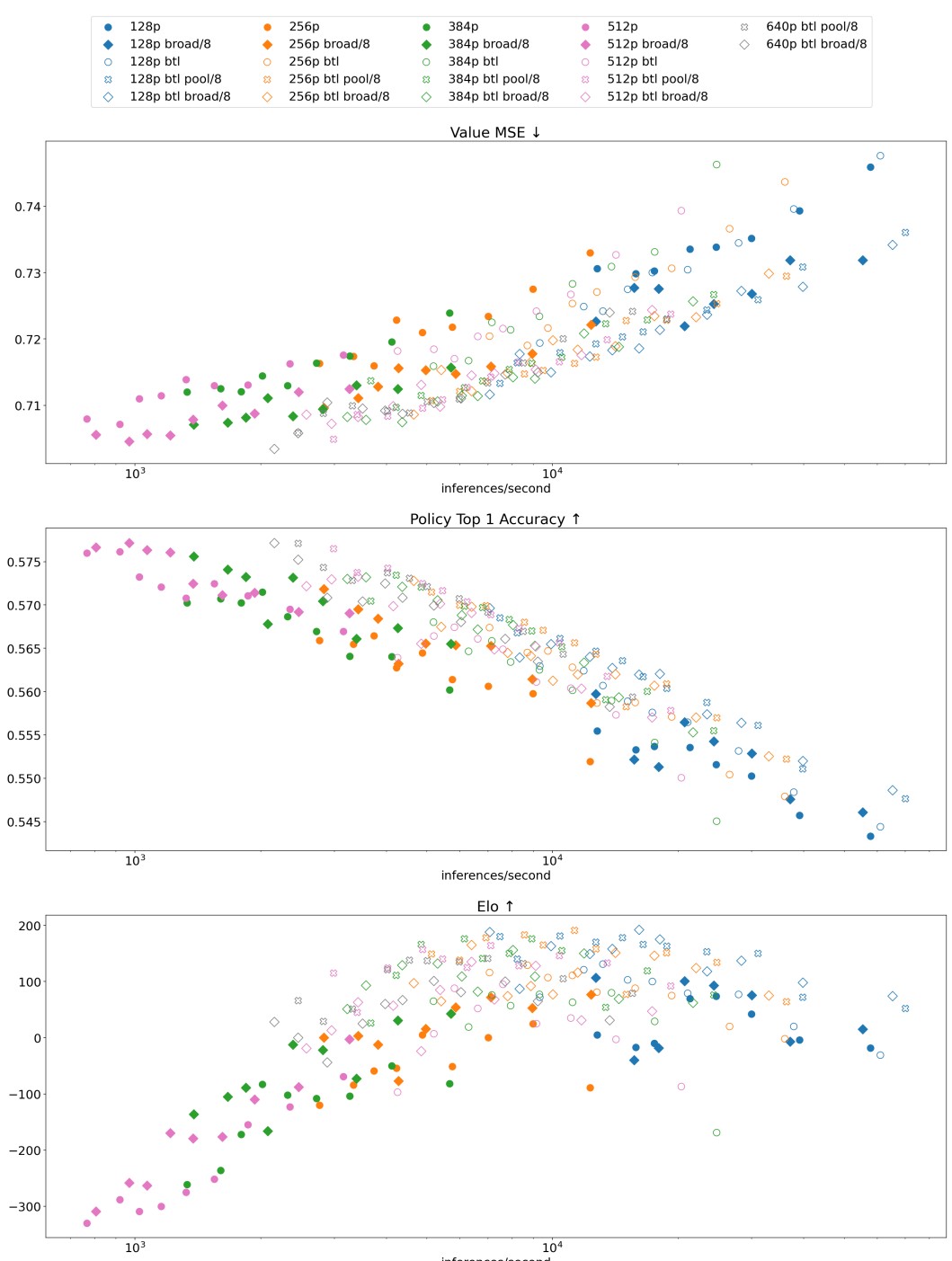

Figure 10: Scaling of value Mean Squared Error (MSE), policy accuracy and playing strength in Elo vs inference speed for different network architectures. In the legend, *btl* indicates bottleneck residual blocks; *broad/8* and *pool/8* indicate broadcast (Figure 11) and pooling blocks (Figure 12) in every 8th layer. *128p*, *256p*, etc. indicate the number of hidden planes.

```python
class BasicBlock(hk.Module):
  """Basic block composed of an inner op, a norm op and a non linearity."""

  def __init__(self, make_inner_op, non_linearity=jax.nn.relu, name='basic'):
    super().__init__(name=name)
    self._op = make_inner_op()
    self._norm = hk.BatchNorm(create_scale=False, create_offset=True, decay_rate=0.999,
                              eps=1e-3)
    self._non_linearity = non_linearity

  def __call__(self, x: Tensor, call_args: CallArgs):
    x = self._op(x)
    x = self._norm(x, is_training=call_args.is_training,
                   test_local_stats=call_args.test_local_stats)
    return  self._non_linearity(x)

class ResBlock(hk.Module):
  r"""Creates a residual block with an optional bottleneck."""

  def __init__(self, stack_size: int, make_first_op, make_inner_op, make_last_op, name):
    super().__init__(name=name)
    assert stack_size >= 2

    self._blocks = []
    for i, make_op in enumerate([make_first_op] + [make_inner_op] *
                                (stack_size - 2) + [make_last_op]):
      self._blocks.append(
          BasicBlock(
              make_inner_op=make_op,
              non_linearity=lambda x: x if i == stack_size - 1 else jax.nn.relu,
              name=f'basic_{i}'))

  def __call__(self, x: Tensor, call_args: CallArgs):
    res = x
    for b in self._blocks:
      res = b(res, call_args)
    return jax.nn.relu(x + res)

class BroadcastResBlock(ResBlock):
  """A residual block that broadcasts information across spatial dimensions.

  The block consists of a sequence of three layers:
   - a layer that mixes information across channels, e.g. a 1x1 convolution.
   - a layer that mixes information within each channel, a dense layer.
   - another layer to mix across channels.

  The same set of weights is used for mixing information within each channel.
  """

  def __init__(self, make_mix_channel_op, name):

    def broadcast(x: jnp.ndarray):
      n, h, w, c = x.shape

      # Process all planes at once, applynig the same linear layer to each.
      x = x.transpose((0, 3, 1, 2))  # NHWC -> NCHW
      x = x.reshape((n, c, h * w))
      x = hk.Linear(h * w, name='broadcast')(x)
      x = jax.nn.relu(x)
      x = x.reshape((n, c, h, w))
      x = x.transpose((0, 2, 3, 1))  # NCHW -> NHWC
      return x

    super().__init__(
        stack_size=3,
        make_first_op=make_mix_channel_op,
        make_inner_op=lambda: broadcast,
        make_last_op=make_mix_channel_op,
        name=name)
```

Figure 11: Bottleneck and broadcast blocks used by the board game network, implemented in JAX (Bradbury et al., 2018) with the neural network library Haiku.

```python
class PoolResBlock(hk.Module):
  """A residual block that broadcasts information across spatial dimensions.

  The block consists of a sequence of three layers:
   - a layer that mixes information across channels, e.g. a 1x1 convolution.
   - a layer that mixes information within each channel, a dense layer.
   - another layer to mix across channels.

  The same set of weights is used for mixing information within each channel.
  """

  def __init__(self,
               make_mix_channel_op: MakeForwardModule,
               name='pool'):
    super().__init__(name=name)
    self._block = functools.partial(
        BasicBlock, make_inner_op=make_mix_channel_op)

  def __call__(self, x: Tensor, call_args: CallArgs):
    a = self._block(non_linearity=jax.nn.relu, name='input_a')(x, call_args)
    b = self._block(non_linearity=jax.nn.relu, name='input_b')(x, call_args)

    b_planes = jnp.concatenate([jnp.mean(b, (1, 2)), jnp.max(b, (1, 2))], -1)
    b_planes = hk.Linear(a.shape[-1], name='mix_channels')(b_planes)
    c = a + b_planes[:, None, None, :]

    x = x + self._block(non_linearity=lambda x: x, name='output')(c, call_args)
    return jax.nn.relu(x)
```

Figure 12: Pooling block, based on Wu (2019).

```python
def make_conv(output_channels: int, kernel_shape: int):
  return functools.partial(hk.Conv2D, output_channels, kernel_shape, with_bias=False,
                           w_init=hk.initializers.TruncatedNormal(0.01))

def make_network(num_layers: int, output_channels: int, bottleneck_channels: int,
                 broadcast_every_n: int):
    blocks = [
        BasicBlock(make_inner_op=make_conv(output_channels), non_linearity=jax.nn.relu,
                   name='init_conv')
    ]

    for i in range(num_layers):
      if broadcast_every_n > 0 and i % broadcast_every_n == broadcast_every_n - 1:
        blocks.append(BroadcastResBlock(
            make_mix_channel_op=make_conv(output_channels, kernel_shape=1),
            name=f'broadcast_{i}'))
      elif bottleneck_channels > 0:
        blocks.append(ResBlock(
            stack_size=4,
            make_first_op=make_conv(bottleneck_channels, kernel_shape=1),
            make_inner_op=make_conv(bottleneck_channels, kernel_shape=3),
            make_last_op=make_conv(output_channels, kernel_shape=1),
            name=f'bottleneck_res_{i}'))
      else:
        blocks.append(ResBlock(
            stack_size=2,
            make_first_op=make_conv(output_channels, kernel_shape=3),
            make_inner_op=make_conv(output_channels, kernel_shape=3),
            make_last_op=make_conv(output_channels, kernel_shape=3),
            name=f'res_{i}'))

    return blocks
```

Figure 13: Usage of modules defined in Figure 11 to create the network used for board game experiments. Alternatively, PoolResBlock can be used instead of BroadcastResBlock.

