# OpenReview forum: "Policy improvement by planning with Gumbel"
_ICLR.cc/2022/Conference — ICLR 2022 Spotlight_

### Official Review · Reviewer_Qbi3 · 2021-10-26

**Correctness:** 4
**Technical Novelty And Significance:** 3
**Empirical Novelty And Significance:** 3
**Recommendation:** 8
**Confidence:** 3

**Main Review:**

## Strengths

* This paper was a breath of fresh air in terms of clarity! Very well written overall. I very much liked the progression from deterministic bandit to stochastic bandit to MCTS.
* I like that the paper emphasizes simple regret instead of cumulative. I completely agree that this is the regret that we should care about in a planning setting.
* The related work is thorough and seems to hit on all the main relevant areas. The authors demonstrate a deep and comprehensive understanding of the literature.
* Though code was not provided, the experimental details look sufficient to reproduce the main results (hardware permitting…)


## Weaknesses/Questions

* I want to better understand what it is about Gumbel specifically that is important here. The paper makes clear that differentiability is not a factor. Instead, Gumbel is used in Algorithm 1 to satisfy Inequality 6. But it seems that the Gumbel distribution itself is not really important -- what’s important to achieve the inequality is that the same $g$ is used both for action selection during search, and for policy improvement after search. Would the same result be achieved in practice by “re-seeding” a random number generator and sampling from the categorical distribution of actions, for example?
* Using completed Q-values to train an improved policy, as described in section 4, makes sense. But if we can do that, don’t we automatically avoid the policy improvement issues exemplified by Example 1, without needing to “plan with Gumbel” as in Algorithm 1? In other words, why can’t we just use Equation 11 (which has nothing to do with Gumbel) instead of using Algorithm 1 to define the improved policy?
* I understand that high variance in action selection could be problematic, and determinism seems like a reasonable hack. But is there any theoretical basis for this?
* I did not understand this sentence: “We cannot use a deterministic action selection at the root node, because we do not remember the visit counts from the same (or similar) state visited in previous episodes.” Could you explain/elaborate?
* The results in Figure 2 use only two random seeds. I know these experiments are very expensive, but two random seeds is just not enough for us to draw meaningful conclusions about statistical significance, so I am interpreting these results with a huge grain of salt. I am glad that 10 seeds were used in Atari.
* The motivation for learning more efficiently from fewer simulations is presumably that with fewer simulations, the overall algorithm runs faster. To what extent is this the case in practice? Approximately how much faster is it to train MuZero with 2 simulations vs 200? Are the simulations the speed bottleneck?
* It is good to see some ablation results, but it would be most ideal if each of the improvements claimed over the original Alpha/MuZero had an associated ablation.

The improvements claimed in the introduction are:
   1. Selecting actions to search (using Algorithm 1 instead of Dirichlet noise)
   2. Selecting actions at the root (using Sequential Halving instead of PUCB)
   3. Selecting actions in the environment (using the Sequential Halving result directly instead of sampling from softmax)
   4. Policy network update (Q-value completion instead of softmax visit count)
   5. Selecting actions (improved policy instead of PUCT)

        The ablations in the paper are:
   * Figure 3a and 6: (d) policy network update
   * Figure 7: (e) selecting actions
   * Figure 8:  (none of the above) other ablations on hyperparameters

        So it would be good to also include ablations for (a), (b), and (c). Let me know if I’ve misaligned anything.

## Minor

   * “we will focus at the action selection” → “we will focus *on* the action selection”
   * In Algorithm 2, is the “Remaining” in the argmax just a singleton set of the best action found by sequential halving? If so, it might be clearer to just say that $A_{n+1}$ is equal to that best action.
   * I happened to already know about Sequential Halving, but it’s likely that a more general audience would not. Maybe add a few sentences explaining it -- the caption in Figure 1 does this to some degree, so just including that description in the main text and saying a bit more would be good.
   * In that spirit, it would be good to give a little more background overall -- for example, clearly stating all of the different subproblems that must be solved in Alpha/MuZero. These subproblems are implicit in the contribution bullet points in the introduction, but they are also not exhaustive.

**Summary Of The Paper:**

This paper considers MCTS with learned search guidance, as in AlphaZero, MuZero, etc. The work proposes several adjustments to the prior works, particularly regarding the way in which actions are selected at the root and non-root nodes, at training and during evaluation; and also the way in which the policy is updated after search. In a simplified bandits setting, the authors point out that the previous method of performing policy updates is not guaranteed to result in a policy improvement, and they propose using a Gumbel reparameterization trick to overcome this limitation. Experiments in Go, Chess, and Atari show that the adjustments are beneficial in the regime of low simulation count.

**Summary Of The Review:**

This paper is likely to be of theoretical and practical interest to many in the ICLR community. My recommendation is not yet higher because I am not convinced of the framing around “planning with Gumbel” and because of the concerns surrounding experiments that I mentioned above.

---

> ### Author Response · Authors · 2021-11-15
> **Response from authors**
>
> Thank you for the uplifting review.
>
> Answers to the questions:
> 1) About Gumbel: Practically, the Gumbel-Top-k trick is an efficient way to implement sampling without replacement. Furthermore, the Gumbel allows us to use the logits again in $\mathrm{argmax}_{a \in \mathcal{A} \mathrm{topn}}(g(a) + \mathrm{logits}(a) + q(a))$
> instead of being limited to
>
> $\mathrm{argmax}_{a \in \mathcal{A} \mathrm{topn}}(q(a))$.
>
> About re-seeding: A method would not produce a policy improvement, if using a different Gumbel $g'$ in $\mathrm{argmax}_{a \in \mathcal{A} \mathrm{topn}}(g'(a) + \mathrm{logits}(a) + q(a))$.
>
> 2) You are absolutely right that Equation 11 is another method to obtain a policy improvement. We hope that the community will use the presented methods for more things. Grill et al. (2020) used a similar policy. In Figure 2, Equation 11 is used by the "MPO MuZero" baseline. The Sequential Halving with Gumbel is still better designed for the minimization of the simple regret.
>
> 3) At a node, the Q-value $q(a)$ can be computed by a stochastic rollout, by a deterministic Q-network, or by another deterministic approximation. We clarified the related sentence.
>
> 4) We added Table 1 with the obtained speedup to Appendix F.
>
> 5) In Figure 2, some of the baselines serve as ablations. For example, the "Replacement" baseline uses sampling with replacement instead of the sampling without replacement. The "MPO MuZero" and MuZero baselines do not use the Sequential Halving with Gumbel. In general, we tried to replace the components with sensible alternatives.

---

> > ### Comment · Reviewer_Qbi3 · 2021-11-29
> > **Thanks for the response**
> >
> > Thank you for addressing my feedback. I have raised my score to 8.

---

### Official Review · Reviewer_dky3 · 2021-10-30

**Correctness:** 4
**Technical Novelty And Significance:** 4
**Empirical Novelty And Significance:** 4
**Recommendation:** 8
**Confidence:** 3

**Main Review:**

This paper proposes to improve AlphaZero and MuZero with the principle of policy improvement by using Gumbel. This paper presents empirical results as well as a theoretical proof. While the method looks interesting especially for the case of using only 2 simulations, some concerns and comments are listed as follows.

First, the intuition of applying Gumbel to AlphaZero/MuZero is not well described. Why would Gumbel distribution outperform the current popular UCT (or PUCT) for AlphaZero/MuZero in some cases?

The authors need to clarify why Equation (7) only needs to consider actions in $A_{topn}$ instead of all actions, while still maintaining policy improvement and satisfying its proof in Appendix. As in (6), the authors state that $E[q(A_{n+1})] \geq \sum_a(\pi(a) * q(a))$ produces a policy improvement, and give Example 1 of Section 3.2, showing that AlphaZero may fail for policy improvement. However, the proposed Algorithm 1 (or 2) may also fail for policy improvement, since in the proposed algorithm the Gumbel method with n = 2 may encounter the same case (even for a simple case g = (1,1,1)). In the current presentation, it is not convincing.

In Algorithm 2, the statement is unclear “Use Sequential Halving with n simulations ...”. Do you mean by recursively using Sequential Halving? Anyway, a lot of messages are missing in “Sequential Halving”, particularly related to any subtree searches (MCTS?) or rollouts (or MuZero) (as in Figure 1). Without knowing this, it is hard to clarify the algorithm. Maybe, the authors should provide the pseudo code for better understanding and comparison, particularly for the whole learning process for both Full Gumbel MuZero and Gumbel MuZero methods.

In the experiments of Go in Section 7.1 and 7.2, the authors anchored to Pachi at 1000 Elo. However, the variance would become high for more than 400 Elo difference, and hence makes Figure 3 and Figure 4 unconvincing. The authors should also evaluate confidence intervals as well, and may consider to use other strong Go programs, e.g., KataGo.

Although Figures 2 and 4 show Elo strength growth, it is more interesting to see if there is any limitation about the strength, i.e., whether the strength will be limited to a certain level (e.g., up to which level when compared with kataGo?). After all, we want to know the limit we can push to, especially for AlphaZero-like methods. Note that I know it would take a huge amount of time for 19x19 Go, however, I think it should work for 9x9.

This paper proposes to redesign the selection for the non-root nodes in Section 5, named “Full Gumbel MuZero”. However, in Figures 4 and 7, Full Gumbel MuZero does not seem to have a clear advantage over Gumbel MuZero. That is, it seems that Gumbel MuZero already worked well without this additional improvement. The authors should give more discussion.

In experiments, more information needs to be shown. For example, the training times, training steps, inferencing times, and the computing resources of the experiments. This is important when using fewer simulations that may significantly reduce these.

Some minor presentation issues are listed as follows.
* In Section 3.1 and in Equation (7), q(a) in [0, 1] should be explicitly defined.
* In Equation (8), is action b the sibling or the children of action a?
* In Equation (11), is sigma the same as the definition in Equation (8)?
* In Section 7.1, for the 9x9 Go, when n is small, is the Sequential Halving applied? Note that the authors state that “Gumbel MuZero with n <= 18 selects without Sequential Halving” only for Atari games (in Section 7.3), but not for Go and Chess.
* In Figure 7 of Appendix C, how does the “Stochastic non-root” select actions?


**Summary Of The Paper:**

This paper redesigns AlphaZero and MuZero with the principle of policy improvement and claims the following. By using the Gumbel-Top-k trick, the completed Q-values, and some other mechanisms, this paper shows the proposed solutions, Gumbel MuZero and Gumbel AlphaZero, learn reliably even with only two simulations, and the experiments in this paper show that the proposed methods match the performance of the original MuZero and AlphaZero.


**Summary Of The Review:**

This paper proposes to improve AlphaZero and MuZero with the principle of policy improvement by using Gumbel. This paper presents empirical results as well as a theoretical proof. While the method looks interesting especially for using only 2 simulations, some comments are concerned as described above.

---

> ### Author Response · Authors · 2021-11-15
> **Response from authors**
>
> Thank you for the detailed review and the explicitly listed concerns. Hopefully our answers will help to resolve the concerns.
>
> 1) UCT and PUCB are not the best choice, because they are designed for different settings. UCT does not use a predictor and needs to visit each action at the start of the search. PUCB was designed to minimize the cumulative regret. At the root of the search tree, we should instead minimize the regret from $A_{n+1}$, i.e., the simple regret. Most importantly, if the number of simulations is smaller than the number of actions, then UCT, PUCB, and AlphaZero PUCT do not guarantee a policy improvement.
>
> 2) We added a detailed proof of the policy improvement to new Appendix B. If applying Algorithm 1 to Example 1, with some Gumbel vectors, the new policy will produce a greater return than the original policy. With other Gumbel vectors, the new policy will produce the same return as the original policy. When computing the expectation, the new policy will produce a greater mean return. Please, tell us if something is still unclear.
>
> 3) Sequential Halving is used only at the root of the search tree. Notice that Figure 1 is explained on a k-armed stochastic bandit. The stochastic bandit has only one state. Figure 1 starts with the set of k actions and progressively makes the set smaller. The original Sequential Halving paper (Karnin et al., 2013) provides a helpful pseudocode. Gumbel MuZero uses unchanged MuZero at non-root subtrees. We are also preparing an open-source release of the Gumbel MuZero and Full Gumbel MuZero search algorithms.
>
> 4) The other reference opponents are different versions of AlphaZero, including AlphaZero trained for twice as many training steps.
>
> 5) We added Table 1 with the obtained speedup to Appendix F. We benefited from the ability to run fast preliminary experiments with a small number of simulations.
>
> Thank you for mentioning the minor presentation issues. We clarified the text.

---

> > ### Author Response · Authors · 2021-11-27
> > **Waiting for a response**
> >
> > Hopefully the added proof is clear. Do not forget to update your rating of the paper correctness.

---

> > > ### Comment · Reviewer_dky3 · 2021-11-27
> > > **Response to Authors**
> > >
> > > Thanks for your clarifications that did address my concerns well. I am happy to change the score (to 8) and other ratings as well. Good paper!

---

### Official Review · Reviewer_Qmbr · 2021-11-01

**Correctness:** 3
**Technical Novelty And Significance:** 3
**Empirical Novelty And Significance:** 4
**Recommendation:** 6
**Confidence:** 4

**Main Review:**

**Primary Strengths**:
1) Several interesting adjustments of how the MCTS operates in AlphaZero/MuZero.
2) Strong results (including interesting ablations in appendices).
3) Majority of paper well-written and clear.

**Primary Weaknesses**:
1) While the majority of the paper is clear, some technical details are (in my opinion) sometimes not clear / only become clear much later on --- including sometimes really important details. See detailed comments below.
2) No mention of whether source code will be made available.

---

**Detailed Comments**:
- At end of page 1, for **Selecting actions to search**, should probably explicitly mention that this Dirichlet noise in AlphaZero is only used in the root (for consistency with some of the subsequent points where you also explicitly make the distinction of root vs. non-root).
- Top of page 2: "We instead propose to select the singular action resulting from the Sequential Halving search procedure." --> I interpreted this as saying that you do not do any exploration at all, purely "greedy" playing, even during the training phase. Only all the way down in Appendix E did it become clear that you do in fact using exploration during training, which contradicts this, so then I assume this actually only refers to evaluation games. This should be clarified.
- In Introduction, the phrase "We instead propose a policy improvement based upon a completion of the root action vales" did not at all paint a picture in my head resembling what actually happens in Section 4. Especially it's difficult to imagine, without reading the entire paper, what *completion of values* would mean. I think I would simplify it and simply say that it's a policy improvement operator based on estimated action values (or advantages). The completion of values (or really, completion of *vectors of values*, i.e. filling in "missing entries") is more of an "annoying" detail necessary to handle situations where some actions didn't get any values, but I don't get the impression that it's actually the core idea; the policy improvement operator still works when there are enough visits to visit every action!
- Under Figure 1, I assume that the $b$ actions in $\max_b N(b)$ that the max iterates over are children / successors / actions of the root node, but this should be made explicit.
- The main paper summarises the proposed policy improvement operator as Eq. (11), and states that a proof for it being a policy improvement operator can be found in Appendix B. However, in Appendix B, I instead find a proof for Algorithm 3, and Algorithm 3 looks very different to Eq. (11). I can... sort of imagine that they might end up working out to the same / a similar thing after the softmax normalises everything into a probability distribution again, but this really doesn't seem obvious. Is it possible to elaborate on this?
- Related to the previous point, I also don't find the intuition suggested by "Intuitively, the completed $Q$-values give zero advantage to the unvisited actions." particularly intuitive. What if all the $q(a)$ values for the visited actions end up being "disappointing", all lower than the expected value for the root based on value network? Surely then the $v_{\pi}$ values would exceed the $q(a)$ values and actually give a positive advantage to unvisited actions? Algorithm 3 in Appendix B does match this intuition much more closely, because only actions that have $q(a)$ values get updated explicitly... although, of course, the subsequent normalisation of the softmax will implicitly then also update unvisited actions again.
- Eq. (14) uses $\pi'(a)$, computed according to Eq. (11), every time the node for which $a$ is an action gets visited. But in between different such visits, its $q(a')$ values (for actions $a'$ that may or may not equal $a$) get updated, so I assume $\pi'(a)$ also gets re-computed to take into account the latest $q$ values? Would be good to explicitly mention this.
- "We cannot use a deterministic action selection at the root node, because we do not remember the visit counts from the same (or similar) state visited in previous episodes." --> I do not understand what this is trying to say. I can certainly understanding a preference for a different action selection method (for example, because we care about simple regret in root, but cumulative regret in non-roots)... but this seems to be saying something else?
- Which agent do you evaluate against in Chess experiments? For Go it was Pachi, but I think that agent doesn't play Chess?
- Appendix C, Eq. (30): please use larger brackets for the big expression after the first fraction. It's rather difficult to read like this, at first I thought there was a missing closing bracket somewhere.

**Other Remarks**:
- I am *very* curious if it would be possible to make this approach work even without a value network $\hat{v}_{\pi}$. It appears to play a relatively "small" role, and already gets mixed with a value estimate based on the children visited so far. What would happen if the value of the policy were estimated solely based on the values backpropagated from previous visits to (other) children? I suppose in practice you would need the value network also to produce the "bandit rewards", but those could be replaced with rollouts... I'm by no means expecting/demanding that large extra experiments get added for this, just curious if the authors happen to already have some insights into this.

**Summary Of The Paper:**

This paper proposes several adjustments to how actions are selected at various stages of the tree searches in AlphaZero/MuZero, with a particular focus on modifications that can significantly improve training performance in training regimes with extremely low iteration budgets per move (even successful training with only 2 iterations per step). Experiments are run using Go, Chess, and two Atari games.

**Summary Of The Review:**

Overall a strong paper, with at this point unfortunately slightly too many places where technical details are not entirely clear or only become clear much later than they should be. Based on the strength of the rest of the paper, I do expect that the authors should be able to clear things up (either in the paper or in comments to my review explaining why I'm wrong) without too much trouble, but I do think it is important to do this before it can be published!

---

> ### Author Response · Authors · 2021-11-15
> **Response from authors**
>
> Thank you for mentioning the unclear points. Your comments helped us to clarify the paper.
>
> Answers to the questions:
> 1) We are preparing an open-source release of the search algorithms.
>
> 2) We updated the proof of the policy improvement with the completed Q-values. The proof is now in Appendix C and the start of the proof mentions the specific instance of Algorithm 3.
> 3) On chess, we evaluate against multiple configurations of Stockfish. Because Stockfish does not benefit from the available TPU, we use the results only to compare AlphaZero and Gumbel AlphaZero.
>
> 4) The search without the value network is an interesting question. The rollouts may be practical, if the episodes are not too long. On Go and chess, we use discount=1, so the rollouts would need to reach the end of the game.

---

### Official Review · Reviewer_urs9 · 2021-11-04

**Correctness:** 4
**Technical Novelty And Significance:** 3
**Empirical Novelty And Significance:** 3
**Recommendation:** 8
**Confidence:** 3

**Main Review:**

- The paper proposes a series of principled improvements to state-of-the-art MCTS planning algorithms. The proposed improvements are particularly effective in low simulation budget settings. The proposed improvements (Gumbel tricks to incorporate the policy network into the tree search, simple regret minimization at the root, planning-learning loop and action selection at non-root nodes) are intuitively clear.
- The details on the planning-learning loop are somewhat sparse. Some of my questions were: What exactly is the update mechanism? Which nodes in the tree search are used to generate training data? These details about the learned policy network seems particularly important given the target setting of very small simulation budgets implying limited lookahead.
- At the smaller simulation budgets, I'd expect the impact of planning to performance is likely limited compared to the learned policy. Is this correct? I'd be curious to better understand how much benefit there is to planning versus simply using the reactive policy without lookahead at the smaller simulation budgets. Please consider including the learned reactive policy as a baseline in the experiments. Some timing information (per decision) would be nice as well to better understand the overhead involved in each method. This might help improve the motivation for planning in small-budget settings, which is currently not described in much detail.
- Overall, the paper contains a number of intuitively clear, principled algorithmic improvements to AlphaZero (and MCTS in general) when used in low budget settings. The learning-planning portion of the paper can be improved with additional detail. The paper is well written and I enjoyed reading it. I think this will make a nice contribution to the literature on anytime planning algorithms with learned components.

UPDATE: I thank the authors for their feedback. After reading it along with the other reviews, I continue to remain positive about the paper.

**Summary Of The Paper:**

The paper proposes a number of principled algorithmic modifications to state-of-the-art planning algorithms (AlphaZero, MuZero) for improving performance in settings with many actions and a relatively small computation and / or sample budget. The main contributions are algorithmic and empirical. The key ideas include the use of the Gumbel-max and top-k tricks along with the use of sequential halving to improve online planning. The paper also proposes a planning-learning loop wherein a policy using the estimated (completed) Q values is learned as well as a different selection policy at non-root nodes. The experiments show that the Gumbel variants of AlphaZero and MuZero perform well in low search budget settings in the domains of Go, Chess and Atari.

**Summary Of The Review:**

Improves anytime planning with a learned component in low simulation budget settings. Could be improved with a bit more detail on the learning-planning loop and additional baselines. Overall, seems like a solid paper to me.

---

> ### Author Response · Authors · 2021-11-15
> **Response from authors**
>
> Thank you for the careful review. We are delighted that you enjoyed reading the paper.
>
> Answers to the questions:
> 1) The other parts of the planning-training loop are the same as in AlphaZero or MuZero. Like in AlphaZero, after finishing a search, the root of the search tree proposes an improved policy as a target for the policy network. To prevent misunderstanding, we are preparing an open-source release of the search algorithms.
>
> 2) We added Figure 9 to show evaluation with different numbers of simulations. Evaluation with 1 simulation corresponds to acting with the reactive policy.
>
> 3) We added Table 1 with the obtained speedup. We benefited from the ability to run fast preliminary experiments with a small number of simulations.

---

### Author Response · Authors · 2021-11-15
**Main changes**

- We are preparing a nice open-source release of the search algorithms.
- We added a policy improvement proof for planning with Gumbel to new Appendix B.
- In Appendix F, we included Table 1 with the obtained speedup.

---

### Public Comment · ~Ivo_Danihelka1 · 2022-03-04
**Open source release**

We prepared an open source library with the new Monte Carlo tree search algorithms:

https://github.com/deepmind/mctx

---

### Decision · Program_Chairs · 2022-01-20

**Decision:**

Accept (Spotlight)

**Comment:**

The paper presents improvements to AlphaZero and MuZero for settings where one is restricted in the number of rollouts. The initial response from reviewers was generally favorable  but the reviewers wanted more details and clarifications of multiple parts of the paper, and further intuition about the Gumbel distribution.  The authors’ responses were detailed and convinced or maintained strong positive support of most reviewers. The authors also stated that they plan to provide a release of the code and also provided a policy improvement proof. Overall this is an interesting approach that is likely to be of significant interest to many.